# Chromosome-level haplotype-resolved assembly of highly heterozygous grass genomes with PhaseGrass

Yutang Chen [1], Dario Copetti [2], Roland Kölliker [1], Martin Mascher [3], Axel Himmelbach [3], Nils Stein [3] & Bruno Studer [1] ✉

Current methods to generate haplotype-resolved assemblies for highly heterozygous genomes suffer from reference bias when relying on reference-based phasing or may generate imbalanced haplomes when relying on graph-based phasing. Here, we report PhaseGrass, an assembly, phasing and scaffolding workflow that is compatible with accurate or error-prone long reads, generating haplotype-resolved assemblies for highly heterozygous genomes. Requiring no parental data, PhaseGrass combines reference-based phasing with haplotype-specific k-mers to partition long reads or unitigs to corresponding haplotypes, thereby avoiding reference bias and uneven haplotype partitioning. Using PhaseGrass, we generated chromosome-level, haplotype-resolved assemblies for the highly heterozygous, allogamous grass species *Lolium perenne* L. and *L. multiflorum* Lam. We find that PhaseGrass can bin 20% more reads to haplotypes than WhatsHap and generated balanced haplomes compared to Hifiasm, which produced largely imbalanced haplomes for *L. multiflorum* due to abundant structural variations between haplotypes. PhaseGrass will facilitate routine generation of haplotype-resolved pangenomes for species with highly heterozygous genomes.

A haplotype-resolved assembly, which correctly captures each of the parental haplotypes, is the best unbiased representation of a heterozygous genome when compared to a haplotype-collapsed assembly[1–3]. With recent advances in long-read sequencing technologies, such as Pacific Biosciences' high-fidelity sequencing (PacBio HiFi) and Oxford Nanopore Technologies (ONT), various methods[1–14] have been developed to generate haplotype-resolved assemblies. One method to enable genome-scale phasing is trio binning[1], which leverages parent-specific k-mers to bin long reads of an $F_1$ offspring into haplotype-specific sets and separately assembles each haploid genome (referred to as haplome hereafter). While advantageous for phasing, the requirement of parental data is a limitation of trio binning. To facilitate chromosome-level phasing without parental data, alternative methods focus on phasing with data only from the same single individual[2,3,5,12].

These so-called single-sample methods, combining local phase information provided by long reads with long-range phase information provided by high-throughput chromosome conformation capture (Hi-C) or strand-specific single-cell sequencing (Strand-seq), have been shown to achieve haplomes with better contiguity and phasing accuracy[2,4,5,12].

Depending on whether a haploid assembly is required for phasing, the single-sample methods can be further divided into two categories: reference-based phasing and graph-based phasing. Reference-based phasing pipelines, such as PGAS[5] and DipAsm[2], rely on a chromosome-level haploid assembly to align long reads and long-range data (for example, Strand-seq or Hi-C) to phase variants called from read alignment. The phased variants determine the haplotype of the aligned long reads covering the variants, and the long reads are partitioned

[1]Molecular Plant Breeding, Institute of Agricultural Sciences, ETH Zurich, Universitaetstrasse 2, 8092 Zurich, Switzerland. [2]Arizona Genomics Institute, School of Plant Sciences, The University of Arizona, Tucson, AZ 85721, USA. [3]Leibniz Institute of Plant Genetics and Crop Plant Research (IPK), Corrensstrasse 3, 06466 Seeland, Germany. ✉e-mail: bruno.studer@usys.ethz.ch

according to their haplotype to assemble each haplome separately. Relying on a haploid assembly for phasing may incur reference bias due to highly divergent genomic regions between haplotypes, resulting in inaccurate long read mapping and partitioning. To avoid reference bias, reference-free, graph-based phasing pipelines, such as Hifiasm[3,4,15] and Verkko[12], were developed, and by integrating long-range data (for example, Hi-C) into the phased assembly graph derived from accurate long reads, telomere-to-telomere haplomes have been achieved[12,15].

However, as indicated by a recent study[15], graph-based phasing may produce imbalanced haplomes with one haplome being much larger than others. This is potentially due to some highly divergent DNA sequences between haplotypes that fail to be aligned during the all-vs-all alignment step of assembly which leads to two independent assembly graphs that are subsequently partitioned to the same haplome. Graph-based phasing needs DNA sequences to be aligned to form one graph with bubbles[4] so that the two sides of the bubbles (alleles or haplotigs) can be partitioned to different haplotypes. A new reference-free tool, HapHiC[14], employing Hi-C data to partition haplotigs, was reported recently to resolve this problem, but, unfortunately, HapHiC relies on highly accurate long reads for assembly, limiting the choice of current long-read sequencing technologies.

To make the most of current long-read sequencing technologies, including both highly accurate and more error-prone long reads (for example, PacBio HiFi and ONT data, respectively), and to generate balanced haplomes, especially for highly heterozygous plant genomes, we developed PhaseGrass. PhaseGrass is a single-sample workflow conducting assembling, phasing and scaffolding with long reads, short reads and long-range data (Hi-C) from the same genome. PhaseGrass can use either PacBio HiFi or ONT data for assembling and phasing, or it can use PacBio HiFi data for assembling and ultra-long ONT data for phasing, integrating the strength of both sequencing technologies. PhaseGrass follows the scheme of reference-based phasing, combining long reads with Hi-C data to achieve chromosome-level phasing. However, PhaseGrass distinguishes itself from current reference-based phasing pipelines[2,5] by binning long reads with haplotype-specific k-mers to avoid reference bias. Using haplotype-specific k-mers has allowed PhaseGrass to bin not only ONT data but also haplotigs derived from PacBio HiFi data, resulting in balanced haplomes. With PhaseGrass, we have generated reference-grade, chromosome-level and haplotype-resolved assemblies for *Lolium perenne* L. and *L. multiflorum* Lam., two of the most important forage and turf grass species with large (2n = 2x = 14, haploid genome size 2.5 Gb[16–18]), repeat-rich (proportion of repeats > 70%[16,17,19]) and highly heterozygous (single nucleotide polymorphism, SNP, frequency of 1 per 20 bp[20]) genomes.

## Results

### PhaseGrass workflow
To generate chromosome-level, haplotype-resolved diploid assemblies, PhaseGrass includes the following steps (Fig. 1): step 1, generating a primary assembly with either ONT or PacBio HiFi data. For ONT data, NextDenovo[21] is chosen to generate the primary assembly, which is then polished with NextPolish[22]. For PacBio HiFi data, Hifiasm[4] is chosen to generate the primary assembly, and the unitigs resulting from Hifiasm are kept for haplotype partitioning in step 4; step 2, generating a chromosome-level, unphased haploid assembly. We developed PurgeGrass to remove redundant allelic contigs from the primary assembly. For genomes with large chromosomes, we recommend using TRITEX[23] for scaffolding, although any other effective scaffolding tool may also be applied. After scaffolding, we provide JuiceGrass, a pipeline consisting of scripts from Juicer[24] and 3D-DNA[25], to generate a Hi-C contact map to conduct manual curation with Juicebox[26]; step 3, reference-based phasing with the chromosome-level, unphased haploid assembly. We developed DipGrass, which takes whole-genome sequencing (WGS) short reads to call SNPs and

then phases SNPs through pseudo-chromosomes using long reads and Hi-C data, resulting in chromosome-level phase blocks; step 4, partitioning of long reads or unitigs to different haplotypes. We developed SortGrass to employ haplotype-specific k-mers extracted from the chromosome-level phase blocks to bin ONT data or unitigs from step 1 to two haplotype-specific sets; step 5, assembling each set of ONT data independently and haplotype-aware polishing of the resulting haplomes. NextDenovo is then used to assemble each haplome, and we provide PolishGrass, a pipeline consisting of NextPolish[22] and Hapo-G[27], to conduct haplotype-aware polishing with long and short reads, respectively. This step is only intended for ONT data; step 6, final scaffolding and manual curation. We developed ScaffoldGrass, a pipeline consisting of YaHS[28] and RagTag[29], to scaffold the two ONT-derived haplomes (from step 5) or the two PacBio HiFi-derived haplomes (from step 4) independently, resulting in two chromosome-level haplomes. We concatenate these two chromosome-level haplomes to construct a diploid Hi-C contact map using JuiceGrass for manual curation, which returns the final chromosome-level haplotype-resolved diploid assembly.

### Chromosome-level, haplotype-resolved assembly with ONT data
We applied PhaseGrass to a highly heterozygous *L. perenne* genotype, DH647 (2n = 2x = 14; estimated haploid genome size ~2.2 Gb; heterozygosity 2.21%, Supplementary Fig. 1), with WGS paired-end short reads, ONT data (mean accuracy QV 11) and Hi-C data (Supplementary Table 1). As a high-quality, chromosome-level haploid assembly of DH647 was already available (Kyuss[30], a doubled haploid progeny derived from anther culture of DH647[31]), we directly used this assembly for referenced-based phasing for DH647. This Kyuss assembly also served as a benchmark for phasing, allowing to evaluate the phasing quality of PhaseGrass by comparing the phase between DH647 haplotypes (generated by PhaseGrass) and Kyuss. Combining the ONT data with the Hi-C data, PhaseGrass phased 99.80% of a total of 6.24 million heterozygous SNPs called based on Kyuss with the short reads of DH647 (Supplementary Table 2). This resulted in seven dense chromosome-level phase blocks of DH647 (Fig. 2a, Supplementary Table 3), accompanied by eight small phase blocks (maximum size 129 kb, Fig. 2a, Supplementary Table 3) observed in regions with low SNP density. These small phase blocks could not be integrated into the chromosome-level phase blocks because no Hi-C phased SNPs were available in these regions (Supplementary Fig. 2a). Comparing the haplotype of DH647 phase blocks with Kyuss suggested that the heterozygous SNPs were correctly phased throughout the pseudo-chromosomes with an estimated phasing error rate of 2% (Fig. 2a). Phase switches were found to be enriched in regions with low SNP density, particularly on Chr1 and Chr4 (Fig. 2a, track 2 and 3), where less heterozygous regions were observed. The large segmental haplotype changes observed in regions with high SNP density at the end of Chr1, Chr3, Chr4, Chr5, and Chr7 (Fig. 2a, track 5, 6, and 7) likely indicate genetic recombination events happening during gamete formation in DH647.

With haplotype-specific k-mers, PhaseGrass binned nearly 50% of the total ONT reads to either haplotype (Fig. 2b), and only 3% of the reads were left unassigned. In contrast, around 23% of the ONT reads were left unassigned by WhatsHap[32], the reference-based phasing tool relying on read alignment to bin long reads (Fig. 2b). This suggested that partitioning long reads with haplotype-specific k-mers can mitigate reference bias (Supplementary Fig. 2b). Assembling each haplotype-specific set of the binned ONT reads independently resulted in two haplomes with a total length of 2.25 and 2.26 Gb and a contig N50 of 1.55 and 1.25 Mb, respectively (Table 1). Around 96% of the contigs were anchored to pseudo-chromosomes in each haplome, resulting in the final haplotype-resolved diploid assembly of DH647 with a scaffold N50 of ~327 Mb (Table 1). The two haplomes of DH647 showed similar assembly sizes (Table 1), BUSCO scores

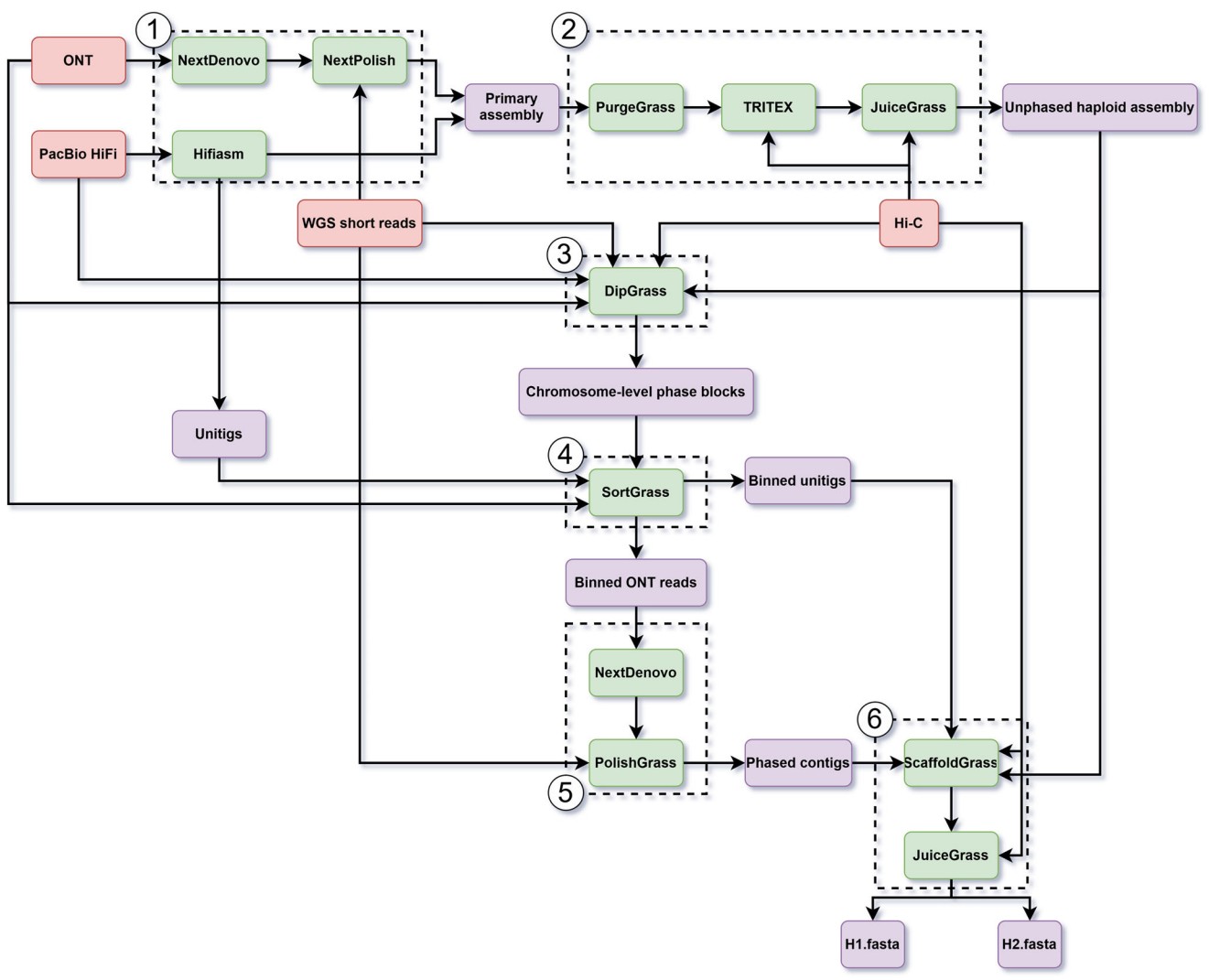

**Fig. 1 | Overview of the PhaseGrass workflow with red, green and purple rectangles indicating input data, computational tools and output results of the workflow, respectively.** PhaseGrass includes the following steps (dashed frames): (1), initial primary contig assembly with either ONT or PacBio HiFi data; (2), generating a chromosome-level, unphased haploid assembly; (3), reference-based phasing; (4), partitioning long reads or unitigs (from step 1) to haplotype-specific sets; (5), assembling each set of binned ONT reads independently and polishing of the resulting contigs. This step is only intended for ONT reads; (6), generating the final chromosome-level haplotype-resolved diploid assembly.

(Table 1), k-mer profiles (Supplementary Fig. 3) and halved ONT read alignment depth (~30-fold, the expected coverage per haplotype, Supplementary Table 1, except for the less heterozygous regions on Chr1 and Chr4 with relatively higher alignment depths, Fig. 2c), suggesting that PhaseGrass correctly partitioned the two haplomes. Aligning each haplome of DH647 against Kyuss, substantially more SNPs were observed with one haplotype but not with the other, and the pattern of recombination between DH647 haplotypes was observed again (Fig. 2a, track 6 and 7), suggesting that PhaseGrass correctly phased the two haplomes. The diploid Hi-C contact map of DH647 and the high genome collinearity between DH647 haplomes and Kyuss suggested that PhaseGrass correctly constructed the pseudo-chromosomes for each haplome of DH647 (Supplementary Fig. 4).

**Chromosome-level, haplotype-resolved assembly with both PacBio HiFi and ONT data**

We also validated PhaseGrass with a more heterozygous *L. multiflorum* genotype, Sikem (2n = 2x = 14, estimated haploid genome size ~2.2 Gb; heterozygosity 3.49%, Supplementary Fig. 1). For Sikem, PacBio HiFi data (read N50 16 kb, mean accuracy QV 33), ONT data (read N50 46 kb, mean accuracy QV 10), Hi-C data and WGS paired-end short

reads were all available (Supplementary table 1). To take advantage of the highly accurate PacBio HiFi data and the ultra-long ONT data, and to compare the phasing results between the two data types, we used PacBio HiFi data for assembly with either PacBio HiFi or ONT data for phasing.

A chromosome-level, unphased haploid assembly containing seven pseudo-chromosomes was first generated for Sikem with the PacBio HiFi and Hi-C data (Supplementary Fig. 5, Table 1). Based on this unphased haploid assembly, we phased heterozygous SNPs called from the WGS short reads with two combinations of data: PacBio HiFi with Hi-C and ONT with Hi-C (referred to as HiFi-Hi-C and ONT-Hi-C hereafter). With either combination, PhaseGrass phased around 99.90% of a total of 6.74 million heterozygous SNPs (Supplementary Table 4). Of these phased SNPs, slightly more SNPs were phased into seven dense chromosome-level phase-blocks by ONT-Hi-C compared to HiFi-Hi-C (99.99% vs 99.67%, Supplementary Table 4). Besides, substantially fewer small phase blocks were obtained with ONT-Hi-C compared to HiFi-Hi-C (46 vs 2564, Fig. 3a, track 4 and 6, Supplementary Data 1). This may be due to the shorter read length of the PacBio HiFi data compared to the ONT data under the same Hi-C sequencing depth (Supplementary Fig. 6), suggesting that longer reads

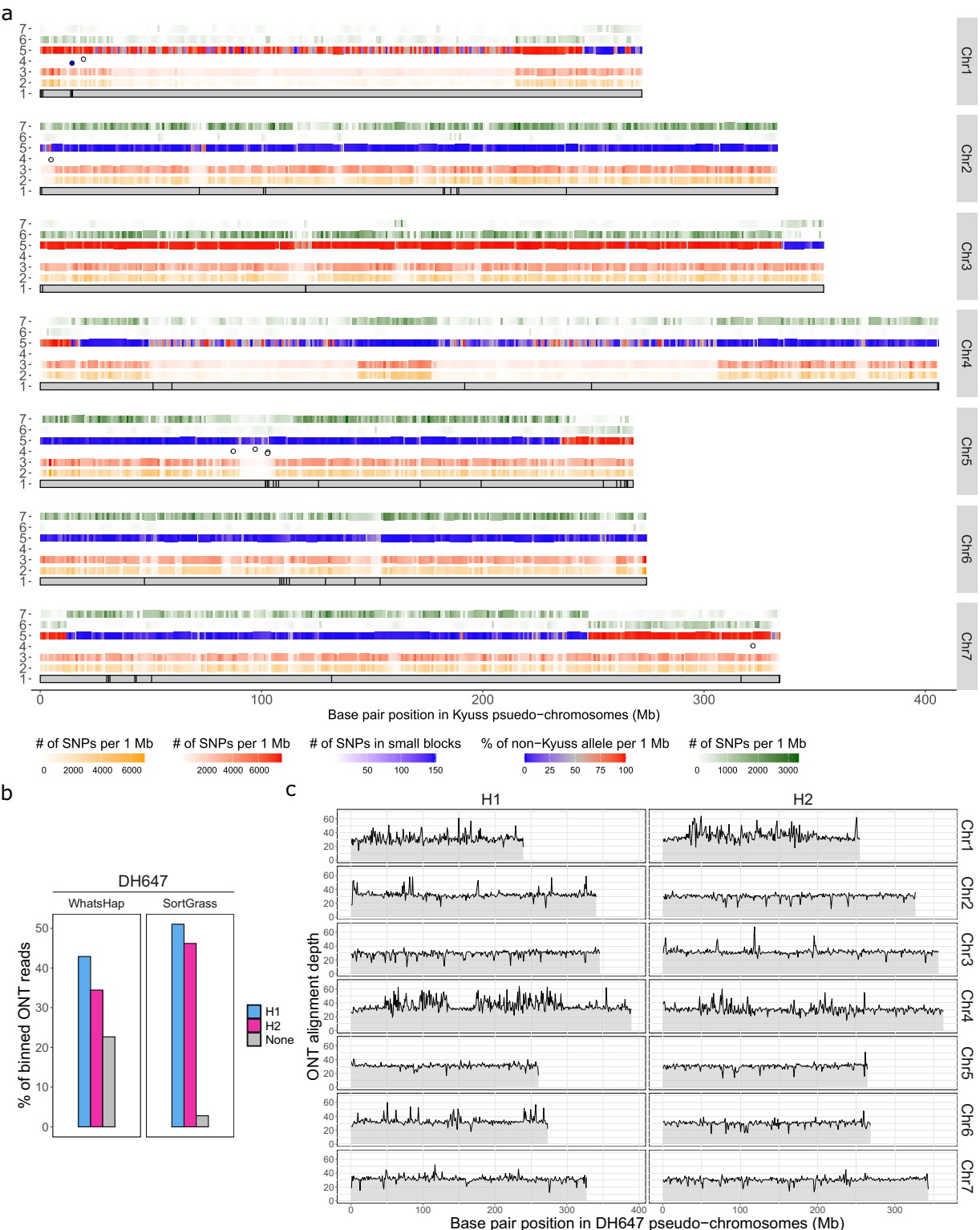

**Fig. 2 | Phasing results for the *Lolium perenne* genotype DH647. a** Reference-based phasing of the DH647 chromosomes (Chr) 1–7: track 1, pseudo-chromosomes of Kyuss, a doubled haploid progeny derived from anther culture of DH647, as reference. Vertical black segments indicate gaps in the Kyuss assembly; track 2, heatmap showing the density of SNPs phased by Hi-C; track 3, heatmap showing the density of all the SNPs phased by Hi-C and long reads in the chromosome-level phase block of DH647; track 4, dots represent small phase blocks that could not be integrated into the chromosome-level phase block; track 5, ratio of non-Kyuss alleles in the DH647 haplotype. Kyuss inherited one haplotype from DH647, and a high (red) or a low (blue) non-Kyuss allele ratio through the pseudo-chromosome suggests correct chromosome-level phasing. A middle ratio (gray) suggests low phasing quality, and the large segmental change of haplotype at the end of some pseudo-chromosomes indicate recombination events; track 6 and 7, heatmaps showing the density of SNPs called from haplome 1 and 2 of DH647 aligned against Kyuss, respectively. **b** Partition of Oxford Nanopore Technologies (ONT) reads from DH647 to haplotype 1 and 2 (blue and purple, respectively) using WhatsHap (left) and SortGrass (right). Gray indicates the percentage of unassigned reads. **c** ONT read alignment coverage in each DH647 haplome (H1 and H2). All DH647 ONT reads were aligned to the diploid assembly of DH647. Source data are provided as a Source Data file.

**Table 1 | Assembly statistics of DH647 and Sikem with PhaseGrass**

| Genome | DH647 | | | Sikem | | |
|---|---|---|---|---|---|---|
| Assembly long read data | ONT | ONT | | PacBio HiFi | PacBio HiFi | |
| Phasing data | unphased | ONT + Hi-C | | unphased | ONT + Hi-C | |
| Assembly | haploid | haplome 1 | haplome 2 | haploid | haplome 1 | haplome 2 |
| Assembly total length (Gb) | 2.26 | 2.25 | 2.26 | 2.50 | 2.37 | 2.40 |
| Contig N50 (Mb)/L50 (#) | 120/7 | 1.55/353 | 1.25/500 | 9.84/82 | 1.13/571 | 1.18/572 |
| Contig N90 (Mb)/L90 (#) | 31.00/20 | 0.27/1801 | 0.27/2080 | 2.09/265 | 0.05/4047 | 0.05/4328 |
| Scaffold N50 (Mb)/L50 (#) | 333/4 | 327/4 | 326/4 | 312/4 | 292/4 | 293/4 |
| Scaffold N90 (Mb)/L90 (#) | 267/7 | 239/7 | 255/7 | 255/7 | 0.05/822 | 0.05/1407 |
| Pseudo-chromosome length (Gb) | 2.24 | 2.18 | 2.18 | 2.28 | 2.00 | 2.01 |
| Unplaced sequences (Mb) | 23 | 73 | 87 | 217 | 371 | 385 |
| Gaps in pseudo-chromosomes (#) | 59 | 3088 | 3,447 | 452 | 3,392 | 3,299 |
| Anchoring rate (%) | 99.12 | 96.89 | 96.46 | 91.20 | 84.39 | 83.75 |
| Base-level accuracy (QV) | 50.00 | 39.19 | 39.19 | 46.79 | 48.90 | 48.90 |
| Total BUSCO (%) | 99.00 | 93.90 | 93.50 | 97.40[a] | 93.70 | 94.20 |
| Duplicated BUSCO (%) | 4.20 | 4.30 | 4.90 | 13.30[a] | 7.30 | 7.40 |

[a]The total and duplicated BUSCO scores shown in the table were the scores of the whole assembly including pseudo-chromosomes and unanchored sequences. Some redundant duplicated allelic sequences were moved to the unanchored sequences during manual curation causing a relatively higher duplicated BUSCO score. Without unanchored sequences, the total and duplicated BUSCO scores of the pseudo-chromosomes were 95.50% and 7.30%, respectively.

can help phase more SNPs into chromosome-level phase blocks even with lower base-level accuracy. Comparing the haplotypes resulting from either combination with the haploid assembly, a similar mosaic pattern was observed with alternate large segmental haplotypes (Fig. 3a, track 7 and 8), suggesting that the two combinations resulted in consistent chromosome-level phasing. The chromosome-level mosaic pattern should not indicate phase switch errors but indicates that the reference haploid assembly was partially phased with a mix of haplotypes. Differences in phasing between the two combinations were mainly enriched locally in regions with low SNP density (Fig. 3a, track 9). This suggested that phase switch errors might be present in these regions, and as there were fewer small phase blocks from ONT-Hi-C in these regions, this might also suggest that longer reads perform better phasing at less heterozygous regions.

With haplotype-specific k-mers resulting from either combination, PhaseGrass evenly binned unitigs (haplotigs with no phasing error[3]) of Sikem to different haplomes (Fig. 3b). In contrast, with Hifiasm, unitigs were unevenly partitioned with a more than 2-fold difference in size between the resulting haplomes (Fig. 3b, Table2). Since ONT-Hi-C resulted in better phasing results, we continued to scaffold the unitigs binned by ONT-Hi-C. This anchored -84% of each set of unitigs to pseudo-chromosomes for each haplome (Table 1), resulting in a chromosome-level, haplotype-resolved diploid assembly for Sikem. The two haplomes were correctly partitioned as suggested by their similar assembly sizes (Table 1), BUSCO scores (Table 1), k-mer profiles (Supplementary Fig. 3) and halved PacBio HiFi read alignment depths (-20-fold, except for some collapsed regions, such as the ~40-fold coverage regions on Chr5, Fig. 3c). Pseudo-chromosomes in both haplomes were correctly constructed as indicated by the diploid Hi-C contact map and the high genome collinearity between these haplomes and Kyuss (Supplementary Fig. 7a, c). Mapping unitigs binned by HiFi-HiC to the chromosome-level haplomes resulted from ONT-Hi-C revealed that differences in phasing between the two combinations led to switches in unitig binning (Supplementary Fig. 7b). Haplotypes from the chromosome-level haplomes were found to be consistent with the haplotypes from a genetic linkage map (Supplementary Fig. 8a), with, on average, 5% alleles in the haplotype being different between the haplome and the genetic linkage map (Supplementary Fig. 8b). This confirmed that PhaseGrass resulted in correct chromosome-level phasing.

HapHiC was applied to Sikem for comparison with PhaseGrass. HapHiC resulted in 14 scaffolds (groups), corresponding to the 14 haplotypes (Supplementary Fig. 9a, Supplementary Table 5). These scaffolds were not partitioned to haplomes, and homologous chromosome pairs were not identified. Compared with HapHiC, PhaseGrass anchored 262 Mb more unitigs to pseudo-chromosomes, showing a higher scaffold N50 (292 Mb vs 263 Mb, Supplementary Fig. 9b).

## High sequence variation between haplotypes causes imbalanced haplomes

To understand why Hifiasm might generate imbalanced haplomes, we investigated the sequence variation between haplotypes by comparing the repetitive and nonrepetitive regions between haplomes. In addition to DH647 and Sikem, we also analyzed a genotype (HEN17)[33] of another allogamous forage crop, red clover (*Trifolium pratense* L.), for which we were able to generate balanced haplomes using Hifiasm (Supplementary Fig. 10, Supplementary Table 6). We found that 80% and 20% of the haplomes for both DH647 and Sikem were repetitive and nonrepetitive, respectively (Supplementary Fig. 11a, Supplementary Table 7). For HEN17, 47% and 53% of the haplomes were found to be repetitive and nonrepetitive, respectively (Supplementary Fig. 11a, Supplementary Table 7). By mapping the repetitive and nonrepetitive sequences from haplome 2 to haplome 1 (Supplementary Table 8), a high median sequence identity of 99% was observed in repetitive regions in DH647, Sikem and HEN17 (Supplementary Fig. 11bc), and a lower median sequence identity of 93%, 80% and 97% was observed in nonrepetitive regions in DH647, Sikem and HEN17, respectively (Fig. 4a, b). This suggested that sequence variation is mainly contributed by nonrepetitive regions between haplotypes, and Sikem showed a much higher sequence variation evenly distributed through haplomes compared to HEN17 (80% vs 97%, Fig. 4a). Given that high sequence variation would cause reads or unitigs between haplotypes not to align[3,4,34], this much higher sequence variation in Sikem was highly likely the reason for Hifiasm generating the imbalanced haplomes.

Further parsing the nonrepetitive sequence alignment resulted in four categories of sequence variation (Fig. 4c, Supplementary Table 9), including single nucleotide variation (SNV), small insertion and deletion (INDEL, length <50 bp), presence and absence variation (PAV, length ≥ 50 bp) and soft-clipped sequence (SC, unaligned bases at the end of a query sequence, and SCs ≥ 50 bp were also considered as PAVs here).

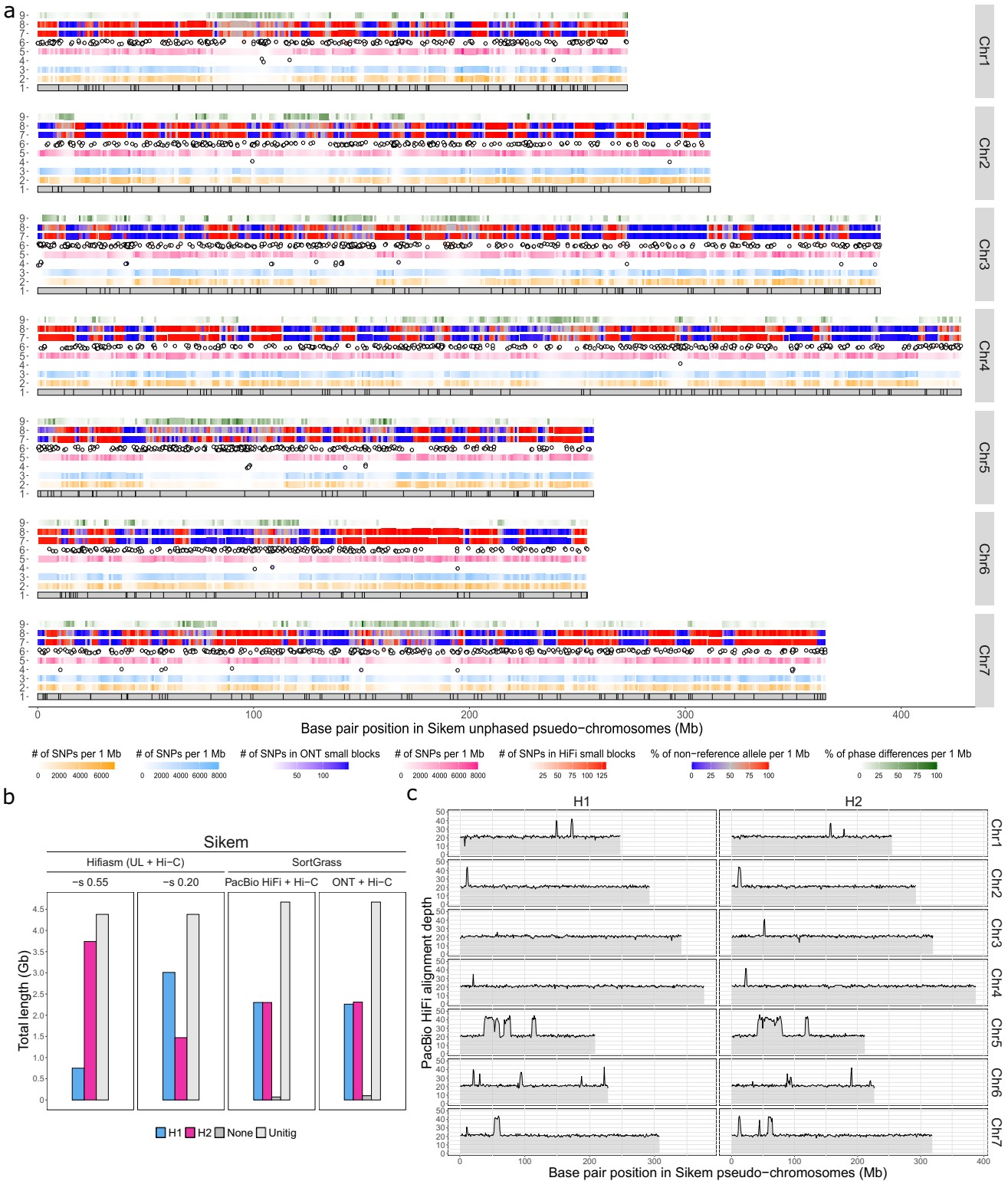

**Fig. 3 | Phasing results for the *Lolium multiflorum* genotype Sikem. a** Reference-based phasing of the Sikem chromosomes (Chr) 1–7 with two combinations of phasing data (ONT-Hi-C and HiFi-Hi-C): track 1, pseudo-chromosomes of the unphased haploid assembly of Sikem as the reference. Vertical black segments indicate gaps in the assembly; track 2, heatmaps showing the density of SNPs phased by Hi-C; track 3 and 5, heatmaps showing phased SNPs in the chromosome-level phase block resulting from ONT-Hi-C and HiFi-Hi-C, respectively; track 4 and 6, dots represent small phase blocks that could be not be integrated into the chromosome-level phase blocks resulting from ONT-Hi-C and HiFi-Hi-C, respectively; track 7 and 8, the ratio of non-reference alleles in the haplotype resulting from ONT-Hi-C and

HiFi-Hi-C, respectively. The low (blue) and high (red) non-reference allele ratio suggests whether the resulting haplotype is the same or different to the reference. A middle value (gray) might suggest phase switches in the resulting haplotype or in the reference; track 9, alleles in the reference that were assigned a different phase between ONT-Hi-C and HiFi-Hi-C, suggesting that ONT and PacBio HiFi data resulted in a different phase for these alleles. **b** Sikem haplotype partition results between Hifiasm and PhaseGrass. Two levels of sequence similarity (-s 0.55 and 0.20) were applied for Hifiasm. **c** PacBio HiFi read alignment coverage in each Sikem haplome (H1 and H2). All Sikem PacBio HiFi reads were aligned to the diploid assembly of Sikem. Source data are provided as a Source Data file.

**Table 2 | Assembly statistics of Sikem with Hifiasm**

| Assembly input data | PacBio HiFi + Hi-C + ultra-long ONT[a] | | | | | |
|---|---|---|---|---|---|---|
| Purge level | −s 0.55 | | | −s 0.20 | | |
| Assembly | utg | H1 | H2 | utg | H1 | H2 |
| Total length (Gb) | 4.38 | 0.75 | 3.74 | 4.38 | 3.01 | 1.47 |
| N50 (Mb) | 11.00 | 24.74 | 29.92 | 11.00 | 49.96 | 26.41 |
| Total BUSCO (%) | 99.40 | 29.80 | 99.50 | 99.40 | 98.90 | 64.70 |
| Duplicated BUSCO (%) | 89.90 | 1.30 | 71.70 | 89.90 | 38.40 | 2.40 |

[a]The ultra-long ONT reads were those longer than 50 kb selected from the total ONT reads.

When compared to HEN17, Sikem showed two times more PAVs within the nonrepetitive regions (9207 vs 4154) and four times more PAVs at the end of the nonrepetitive regions (SCs ≥ 50 bp, 21,995 vs 4877, Supplementary Table 9). Compared to other variations, SCs contributed more to the low sequence identity (Fig. 4c), suggesting that PAVs underlying the high sequence variation between haplotypes in Sikem were likely the major factor causing Hifiasm to generate imbalanced haplomes.

## Discussion

Despite the rapid advance in long-read sequencing technologies, generating haplotype-resolved assemblies for large, repeat-rich and highly heterozygous genomes is still challenging. For example, in this study, Hifiasm produced largely imbalanced haplomes for the highly heterozygous *L. multiflorum* genotype Sikem. To overcome limitations of current phasing pipelines[1,15,32], we provide PhaseGrass, an assembly, phasing and scaffolding workflow for highly heterozygous genomes. With PhaseGrass, we have generated chromosome-level, haplotype-resolved assemblies for two highly heterozygous allogamous grasses species (*L. perenne* and *L. multiflorum*), and based on the assemblies, we have shown several advantages of PhaseGrass.

First, PhaseGrass requires no parental data, but it can achieve accurate chromosome-level phasing with long reads and Hi-C data. This has been validated by the consistency of haplotypes between DH647 (generated by PhaseGrass) and Kyuss, which is a doubled haploid progeny derived from DH647 sharing one of the two haplotypes from DH647 with only occasional recombination events (Fig. 2a). The accuracy of chromosome-level phasing has also been validated by the consistency of alleles between haplotypes of Sikem generated by PhaseGrass and haplotypes from a genetic linkage map of Sikem, with 5% alleles on average being different (Supplementary Fig. 8).

Second, PhaseGrass improves long-read or unitig binning. PhaseGrass leverages haplotype-specific k-mers resulted from phasing to partition long reads or unitigs to haplotype-specific sets. This enabled PhaseGrass to mitigate reference bias (Supplementary Fig. 2) to assign 20% more ONT reads to haplotypes compared to WhatsHap[32] (Fig. 2b). For highly divergent regions with large presence and absence variations, reference bias might lead only long reads sharing the same haplotype with the reference to be aligned to the assembly. Thus, less reads could be assigned to haplotypes by WhatsHap. In contrast, it is easier to match short k-mers (21 bp) with sequences from highly divergent regions. Thus, k-mers can improve long read or unitig partition. Given the effect of reference bias, calling SNPs from the alignment of short reads is advantageous for reference-based phasing. This is because, if only long reads that share the same haplotype as the reference can be aligned to the assembly, then no variation between those long reads and the assembly would be detectable.

The haplotype-specific k-mers also allowed PhaseGrass to bin haplotigs more evenly compared to Hifiasm[15], from which highly imbalanced haplomes were obtained for Sikem with one haplome being two times larger than the other (Fig. 3b).

Third, PhaseGrass is compatible with both ONT and PacBio HiFi data. State-of-the-art phasing pipelines, such as Hifiasm[15] (ONT R9

reads not supported), Verkko[12], HapHiC[14], and TRITEX[23,35], all require accurate long reads for assembly with optional less accurate ONT data for long-range phasing or resolving repeats[12,15]. PhaseGrass differs itself from these pipelines with its capability of generating haplotype-resolved assemblies with only ONT data (for example, the DH647 assembly). Besides, with Sikem, PhaseGrass showed that it can also take advantage of both data types, using accurate PacBio HiFi data for assembly and using ultra-long ONT data for phasing.

By comparing Sikem with HEN17, we found that abundant PAVs in nonrepetitive regions (Fig. 4) might be the reason for Hifiasm[15] generating imbalanced haplomes. This provides important insights into abundant sequence variation between haplotypes of complex plant genomes and may help improve graph-based phasing methods, especially when applying these methods to highly heterozygous genomes beyond plant species.

Currently, PhaseGrass can only be applied to diploid species with high heterozygosity uniformly distributed through genomes. More work is needed to extend PhaseGrass to phase polyploid species. Nevertheless, with all its advantages, PhaseGrass will be a valuable tool to enable routine haplotype-resolved pangenomes to advance research and breeding for species with complex genomes[36,37].

## Methods
### PhaseGrass workflow

PhaseGrass is an assembly, phasing and scaffolding workflow, compatible with both ONT and PacBio HiFi data, generating chromosome-level, haplotype-resolved assemblies for highly heterozygous genomes. PhaseGrass is a single-sample reference-based method leveraging haplotype-specific k-mers to bin not only ONT reads but also unitigs derived from PacBio HiFi reads to different haplotypes. Combining reference-based phasing with haplotype-specific k-mers avoids reference bias and the uneven partition of haplotypes observed with graph-based phasing method (Hifiasm[3,4,15]). An overview of the PhaseGrass workflow is illustrated by a flowchart generated with drawio[38] (Fig. 1), and each step in the workflow is described in detail below:

Step 1: generating a primary assembly with either ONT or PacBio HiFi data.

To generate a primary haploid assembly, we use NextDenovo[21] and Hifiasm (-l0 and -u0) for ONT and PacBio HiFi reads, respectively. For the NextDenovo-based ONT read assembly, NextPolish[22] is used to polish the primary assembly with ONT long reads and WGS short reads. Alternative assemblers can be used according to the user's preference or experience.

Step 2: generating a chromosome-level, unphased haploid assembly.

Based on a pilot study with another *L. multiflorum* genotype from a Swiss cultivar 'Rabiosa' (hereafter referred to as Rabiosa), we have seen the potential of purging more redundant haplotigs using micro-synteny and BUSCO genes[39]. Therefore, to more effectively purge redundant haplotigs from the primary assembly generated in Step 1, we developed PurgeGrass, a pipeline that leverages all-by-all alignment, micro-synteny and BUSCO genes to pair allelic contigs and remove redundant allelic contigs. Pair-wise all-by-all alignment

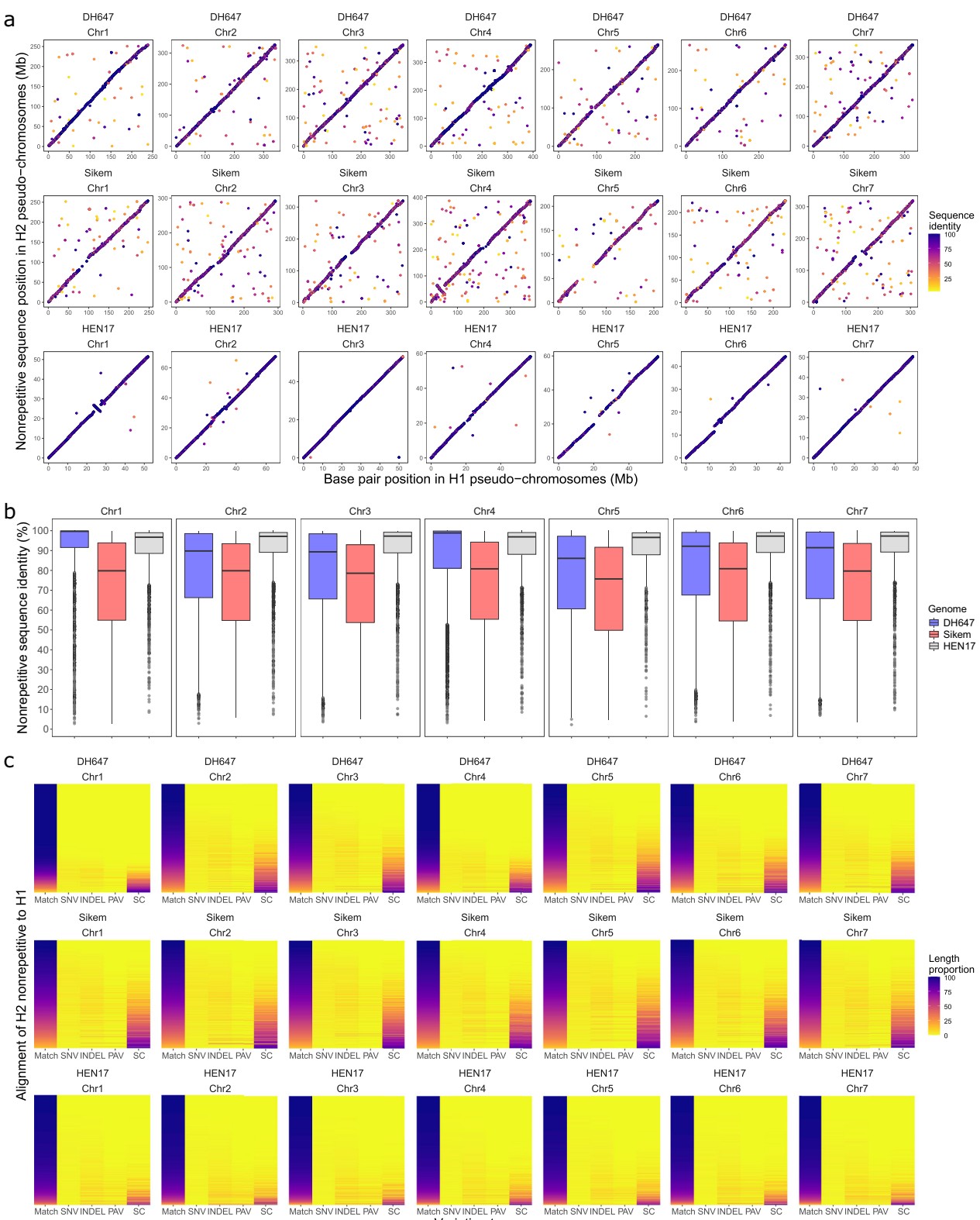

**Fig. 4 | Comparison of nonrepetitive regions between haplomes of the *Lolium perenne* genotype DH647, the *L. multiflorum* genotype Sikem and the *Trifolium pratense* genotype HEN17. a** Intra-chromosome dot plot of nonrepetitive sequences between haplome 1 (H1) and haplome 2 (H2). Each dot represents one alignment (mapping quality value > 0 and query length ≥ 2 kb), and color corresponds to the alignment sequence identity. **b** Distribution of sequence identity of nonrepetitive sequences alignment between haplomes within chromosomes. Boxplots show median (black line), first and third quartile (hinges), 1.5 × inter-quartile-range (whiskers) and outliers (dots). **c** Intra-chromosome sequence variation in nonrepetitive regions between haplomes. In every heatmap, each row represents

one alignment between haplomes. Each alignment is parsed to four types of variation, including single nucleotide variation (SNV), small insertion and deletion (INDEL, length <50 bp), presence and absence variation (PAV, length ≥ 50 bp) and soft-clipped sequence (SC, unaligned bases at the end of a query sequence, and SCs ≥ 50 bp are also considered as PAVs). Matched bases (Match) in the alignment indicates the alignment sequence identity. Each column is colored according to its length in proportion to the total alignment length. From top to bottom, alignments (rows) were sorted by sequence identity from high to low. Source data are provided as a Source Data file.

between contigs is performed by Purge Haplotigs[40]. Micro-synteny between allelic contigs is detected using MCScanX[41] with transcripts from a previous study[18] which are mapped to the contigs by GMAP[42], and contigs sharing more than five genes in synteny are considered as a pair of allelic contigs. BUSCO genes are detected by BUSCO[43] (v5.3.2 with database embryophyta_odb10). Two contigs would be considered a pair of allelic contigs if they share at least one BUSCO gene. The three lines of evidence of allelic contigs together with the primary assembly are fed into PurgeGrass to generate the deduplicated haploid assembly. BUSCO genes in the deduplicated haploid assembly are detected using BUSCO to check remaining duplication. A duplicated BUSCO score lower than 10% together with a small loss in total BUSCO score (<5%) indicates a successful purge of haplotigs.

To scaffold the deduplicated haploid assembly, we use the TRITEX[23] pipeline, which leverages a reference (or a genetic linkage map) and Hi-C data to construct pseudo-chromosomes. For Sikem, Kyuss v2.0[30] was used as the reference to scaffold its deduplicated haploid assembly. Any other favored scaffolding pipelines can be chosen by the user. Here, we chose TRITEX because it performs well in scaffolding large chromosomes as for wheat (*Triticum aestivum* L.)[23], barley (*Hordeum vulgare* L.)[44,45], rye (*Secale cereale* L.)[46], oat (*Avena sativa* L.)[47] and faba bean (*Vicia faba* L.)[48].

Manual curation is conducted at a high resolution with JuiceBox[26] for the pseudo-chromosomes generated by TRITEX. We developed JuiceGrass, a pipeline implemented with Snakemake[49], employing Juicer[24] and 3D-DNA[25] to generate the Hi-C contact map for the pseudo-chromosomes. The Hi-C contact map is loaded into JuiceBox for manual curation. During manual curation, remaining redundant allelic sequences overlooked by PurgeGrass can be removed based on the Hi-C contact map, following the suggestion from Extended Data Fig. 6 of Arang et al.[50]. After manual curation, the final chromosome-level unphased haploid assembly is generated with a script (post_JBT) accompanied by JuiceGrasus.

Step 3: reference-based phasing with the chromosome-level, unphased haploid assembly.

We developed the reference-based phasing pipeline DipGrass (implemented using Snakemake), which takes long reads (either ONT or PacBio HiFi), WGS short reads, Hi-C data and the unphased haploid assembly as input and outputs a VCF file with phased SNPs. DipGrass performs the following tasks: Short reads are mapped to the pseudo-chromosomes of the unphased haploid assembly using BWA-MEM[51] and SAMTools[52], and then SNPs are called with freebayes-parallel[53]. Heterozygous SNPs are selected using BCFTools[52] with user-defined criteria (mapping quality ≥ 20 and mapping depth ≥ 10) for further phasing. Long-range phasing for SNPs with Hi-C data is accomplished using HapCUT2[54], and the longest phase block per pseudo-chromosome (in our case, the longest phase block spanned the whole pseudo-chromosome) is used to provide the long-range phase information for later phasing with long reads. Long reads are then mapped to the pseudo-chromosomes of the unphased haploid assembly using Winnowmap2[55]. The local phase information from the long-read alignment together with the long-rage phase information from the Hi-C mapping is fed into WhatsHap[32] to phase all the heterozygous SNPs selected, resulting in the final phased VCF file containing chromosome-level phase blocks with dense SNPs.

Step 4: partitioning long reads or unitigs to different haplotypes.

We developed SortGrass, a pipeline implemented with Snakemake, to partition long reads or unitigs to different haplotypes with haplotype-specific k-mers. SortGrass takes the phased VCF file (from step 3, DipGrass), the unphased haploid assembly (from step 2) and the ONT reads or the unitigs (from step 1) as input, and it bins ONT reads or unitigs to three categories: haplotype 1, haplotype 2 and unassigned. SortGrass performs the following tasks: First, with the phased SNPs constituting the chromosome-level phase blocks in the VCF file and the pseudo-chromosomes of the unphased haploid assembly, two

haplotype sequences per pseudo-chromosome are generated using BCFTools consensus. Next, one set of 21-mers is extracted from each haplotype sequence using KMC[56], and common k-mers between haplotypes are removed from each set of 21-mers using kmc_tools (subtract) to obtain the haplotype-specific 21-mers. With the haplotype-specific 21-mers, trio_binning[57] is used to classify ONT reads or unitigs to different haplotypes. After classification, a text file with names of reads or unitgs and their associated haplotypes is generated. Finally, Seqkit[58] is used to extract ONT reads or unitgs for each haplotype with the corresponding names in the text file, resulting in the three output sequence files including haplotype 1, haplotype2 and unassigned ONT reads or unitigs.

Step 5: assembling each set of ONT reads independently and haplotype-aware polishing of the resulting haplomes.

This step is only intended when ONT reads are used for assembly. The unassigned reads are concatenated with each set of haplotype reads for assembly using NextDenovo, resulting in two sets of contigs. To polish the two resulting sets of contigs, we developed a pipeline named PolishGrass, implemented with Snakemake. PolishGrass performs the following tasks: First, each set of contigs is polished separately for two rounds using NextPolish with the corresponding reads used for assembly. After long read polishing, the two sets of polished contigs are concatenated to one diploid assembly. Then the diploid assembly is polished with short reads for two rounds using a haplotype-aware polisher, Hapo-G[27], resulting in the final two sets of polished contigs.

Step 6: final scaffolding and manual curation.

We developed ScaffoldGrass, a pipeline implemented with Snakemake, to scaffold the two sets of contigs generated with ONT reads or the two sets of unitigs derived from PacBio HiFi reads. ScaffoldGrass first uses YaHS[28] to scaffold each set of contigs or unitigs with Hi-C data, resulting in two sets of scaffolds. ScaffoldGrass then applies RagTag[29] to anchor each set of scaffolds to pseudo-chromosomes with the help of the chromosome-level, unphased haploid assembly or the reference used for DipGrass. ScaffoldGrass concatenates the resulting two sets of scaffolds from RagTag and outputs a diploid assembly. A Hi-C contact map is constructed (with Hi-C read mapping quality >0) using JuiceGrass for the diploid assembly, and manual curation is conducted with the Hi-C contact map using JuiceBox. During manual curation, sequences that have much stronger Hi-C contact with the other haplome are likely to be partitioned to the wrong haplome by SortGrass. These sequences can be manually moved to the correct haplome. After manual curation, the final diploid assembly with the two haplomes is generated with post_JBT (accompanied by JuiceGrass). For genomes with a relatively low heterozygosity, mapping Hi-C reads to the diploid assembly might result in many ambiguous alignments (mapping quality = 0). In these cases, we provide another script (ScaffoldGrass_hap), which maps all Hi-C reads to each haplome and follows the same method of ScaffoldGrass (with Hi-C read mapping quality ≥30) to scaffold each haplome separately.

## Evaluating phasing quality of DH647 based on a haploid progeny genome

To evaluate the phasing quality of reference-based phasing (implemented in DipGrass of PhaseGrass) for DH647, we compared the alleles of the phased SNPs of DH647 with the alleles in the reference (Kyuss v2.0, the doubled haploid progeny derived from anther culture of DH647[31]). If the SNPs of DH647 are correctly phased, then one of the two haplotypes derived from these phased SNPs should be the same haplotype (except for occasional segmental phase switches due to recombination events in DH647 during gamete formation) since Kyuss inherited one of the two homologous chromosomes/haplotypes from DH647. Additionally, to evaluate the phasing quality in the final haplomes, the two haplomes of DH647 were aligned to Kyuss to call SNPs. A clearly higher number of SNPs would be called with one

haplotype compared to the other if the phasing quality of the final haplomes is high.

The evaluation results were visualized as Fig. 2a using R[59] with input data from the following sources: The length of pseudo-chromosomes of Kyuss and the position of gaps in the pseudo-chromosomes (Fig. 2a, track 1) were obtained from Kyuss v2.0. The Hi-C phased SNPs (Fig. 2a, track 2) were obtained from HapCUT2 with DipGrass. The total phased SNPs in the chromosome-level phase blocks (Fig. 2a, track 3) were derived from the phased VCF file generated by WhatsHap using DipGrass with ONT and Hi-C data. Phase block statistics (Fig. 2a, track 4) were generated by WhatsHap stats with DipGrass. The ratio of non-Kyuss allele per 1 Mb window (Fig. 2a, track 5) was calculated using an R script by counting the number of non-Kyuss alleles in one haplotype of DH647 in the phased VCF file generated by WhatsHap. From one end to the other, one haplotype of DH647 should be the same or opposite to Kyuss (except in the case of recombination), as Kyuss was derived from anther culture of DH647. Therefore, the ratio of non-Kyuss allele in one DH647 haplotype per 1 Mb window could reflect the phasing accuracy. Since Chr2 and Chr6 showed no recombination and no large homozygous regions, a median value of the ratio of non-Kyuss allele was calculated with all the 1 Mb windows through the chromosomes, and this median value was taken to represent the phase switch rate in DH647 haplotypes. SNPs (Fig. 2a, track 6 and 7) were called using MUMmer[60] (show-snps) based on the whole chromosome alignment between DH647 pseudo-chromosomes and Kyuss pseudo-chromosomes. To more efficiently complete whole chromosome alignment, we split DH647 pseudo-chromosomes into 100 kb windows using SeqKit (seqkit sliding) and then aligned these windows to Kyuss pseudo-chromosomes using minimap2[61]. The resulting alignment file was converted to delta format using RagTag (paf2delta), and SNPs were called from the delta alignment file using MUMmer (show-snps).

## Comparing read partition between PhaseGrass and WhatsHap using DH647 ONT data

The total ONT reads of DH647, based on their corresponding haplotype assigned by WhatsHap (whatshap split, implemented in DipGrass) and by SortGrass, respectively, were split into three files (haplotype 1, haplotype 2 and unassigned). Statistics of reads in the three files were calculated using SeqKit (seqkit stats). The read statistics between WhatsHap and SortGrass were compared and visualized with R (Fig. 2b).

## Evaluating phasing quality of Sikem based on its haploid assembly

Following the same approach to evaluate the phasing quality as used for DH647, we compared the alleles of the phased SNPs resulting from DipGrass with the unphased haploid assembly of Sikem, which was used as the reference for reference-based phasing with DipGrass. Since the reference here was a mosaic sequence assembly with alleles alternating between the two haplotypes of Sikem, the comparison between the alleles of the phased SNPs and the corresponding reference alleles may only indirectly indicate the phasing quality. As the reference is mosaic, an alternating pattern of haplotypes would be expected for haplotypes derived from the phased SNPs when compared with the reference. Additionally, phasing quality between different sequencing technologies was also compared (ONT vs PacBio HiFi) based on the haploid assembly.

The results of the comparison were visualized in Fig. 3a using R with input data from the following sources: The length of the pseudo-chromosomes of the Sikem unphased haploid assembly (Fig. 3a, track 1) was obtained using SAMTools (samtools faidx). The position of gaps in the pseudo-chromosomes (Fig. 3a, track 1) was obtained using SeqKit (seqkit locate -p N+). The unphased haploid assembly of Sikem

was generated with PhaseGrass following methods described in PhaseGrass step 1 to 3. Track 2, the Hi-C phased SNPs were obtained from HapCUT2 with DipGrass. The total phased SNPs in the chromosome-level phase blocks (Fig. 3a, track 3 and 5) were derived from the phased VCF file generated by WhatsHap with DipGrass using two combinations of data: (1) ONT with Hi-C (ONT-Hi-C) and (2) PacBio HiFi with Hi-C (HiFi-Hi-C), respectively. The statistics of small phase blocks (Fig. 3a, track 4 and 6) were generated by WhatsHap with DipGrass. The ratio of non-reference allele per 1 Mb (Fig. 3a, track 7 and 8) was calculated using an R script by counting the number of non-reference alleles in one haplotype of Sikem in the phased VCF file generated by WhatsHap with DipGrass. Since the reference is a mosaic sequence with large segmental phase switches, track 7 and 8 can only suggest the local phasing quality. The number of different alleles between haplotypes generated with ONT-Hi-C and HiFi-Hi-C (Fig. 3a, track 9) were counted with an R script.

## Evaluating global phasing quality with a genetic linkage map of Sikem

Track 7 and 8 in Fig. 3a can only indicate the local phasing quality of PhaseGrass. To assess the global phasing quality, we compared the haplotype of Sikem generated by PhaseGrass with the haplotype from a genetic linkage map of Sikem. The genetic linkage map of Sikem was generated using LepMAP3[62] with genotyping-by-sequencing (GBS) data obtained from 305 F1 individuals derived from a cross between Sikem and Rabiosa[39]. GBS data were mapped to each haplome of Sikem generated by PhaseGrass using BWA-MEM, and SNPs were called using the pipeline suggested by LepMAP3 (https://sourceforge.net/p/lep-map3/wiki/LM3%20Home/#sequencing-data-processing-pipeline).

Only with this SNP calling pipeline, LepMAP3 can output the phased alleles (haplotypes) in the resulting genetic linkage map. Alleles in the haplomes generated by PhaseGrass and alleles in the haplotypes from the genetic linkage map were compared with an R script (Supplementary Fig. 8).

## Comparing unitig partition between PhaseGrass and Hifiasm using Sikem PacBio HiFi data

Two methods were compared on the partitioning of unitigs: Hifiasm (UL + Hi-C) and SortGrass. Ultra-long ONT reads were extracted from the total ONT reads of Sikem using SeqKit (seqkit seq -m 50000) with a minimum read length of 50 kb. PacBio HiFi data, ultra-long ONT reads and Hi-C data were fed into Hifiasm (0.19.8-r603) for assembly. Two levels of similarity threshold (−s 0.20 and −s 0.55) for purging duplicated haplotigs were applied with Hifiasm. Hifiasm partitioned unitigs to two haplotype contig assemblies (h1 and h2). The total length of the two haplotype contig assemblies and the unitig assembly was calculated using assembly-stats[63]. For SortGrass, the input unitig assembly was generated using Hifiasm (-l0 -u0) with only PacBio HiFi data (PhaseGrass, step 1), and the unitigs were binned to three files: haplotype 1, haplotype 2 and unassigned. The total length of the three binned unitig files and the whole unitig assembly was calculated using assembly-stats. The total length of the assemblies from the two methods was compared and visualized with R (Fig. 3b).

## Comparing unitig partition between different combinations of data

As aforementioned, Sikem was phased with two combinations of data, namely ONT-Hi-C and HiFi-Hi-C. The difference in phasing between the two combinations is shown in track 9 of Fig. 3a. To check the difference in unitig partition between the two combinations, we concatenated the unitigs (haplotype 1 and 2) binned by HiFi-Hi-C and aligned the concatenated unitigs to the chromosome-level haplotype-resolved diploid assembly (phased by ONT-Hi-C) using minimap2. The alignments were visualized with an R script, and only primary alignments within the pseudo-chromosomes with a length equal to or greater than 100 bp

and a mapping quality score equal to or greater than 50 were visualized (Supplementary Fig. 7b).

## Assembly quality check

Basic assembly statistics, such as assembly size, N50/L50 and N90/L90, were calculated using assembly-stats (Tables 1 and 2). The number of gaps, the length of pseudo-chromosomes and the length of unanchored sequences were calculated using SeqKit (Table 1). The assembly anchoring rate was calculated by dividing the length of the unanchored sequences to the total length of the assembly (Table 1). Base-level accuracy of the assemblies was calculated using POLCA[64] based on the alignment of WGS short reads to the assemblies (Table 1). BUSCO scores were obtained by applying BUSCO (v5.3.2 with database embryophyta_odb10) to the assemblies (Tables 1 and 2). Long read alignment depth of the assemblies was calculated using mosdepth[65] with the total long reads mapped to the diploid assembly using minimap2 (Figs. 2c and 3c). K-mer profile of the assemblies were generated using KAT[66] with WGS short reads (Supplementary Fig. 3).

## Evaluating pseudo-chromosome structure correctness

The structural correctness of the pseudo-chromosomes was evaluated by the Hi-C contact map and the genome synteny between the haplome and the high-quality reference (Kyuss v2.0)[23,30]. The diploid Hi-C contact map was generated using JuiceGrass with Hi-C reads aligned to the chromosome-level haplotype-resolved diploid assembly (Supplementary Fig. 4ab and Supplementary Fig. 7a). For the unphased haploid assembly of Sikem, the Hi-C contact map was generated with JuiceGrass by applying Hi-C reads to the haploid assembly (Supplementary Fig. 5a). Genome synteny between haplomes and Kyuss pseudo-chromosomes (Supplementary Fig. 4c and Supplementary Fig. 7c) was visualized using R with nonrepetitive sequences in the haplome aligned to Kyuss. A custom pipeline was developed to accomplish the alignment of nonrepetitive sequences to the reference, and it performed the following tasks: First, repetitive sequences in the haplome (either haplotype 1 or 2) were detected based on k-mer coverage using KAT sect. Comparing the haplome with itself, any regions in the haplome with a k-mer coverage equal to or greater than 2 were considered as repetitive. Given the presence of gaps (N) in the haplome, the repetitive regions were first masked to N using BEDTools[67], and then any unmasked sequences in the haplome were extracted as nonrepetitive sequences using SeqKit. The nonrepetitive sequences were mapped to Kyuss using minimap2. Alignments with a length equal to or greater than 2 kb and a mapping quality score equal to or greater than 60 were visualized. The synteny between Sikem unphased haploid assembly and Kyuss was generated with the same method (Supplementary Fig. 5b).

## Haplotype-resolved genome assembly of HEN17

We used Hifiasm (Hi-C) to generate a haplotype-resolved assembly for HEN17 with PacBio HiFi and Hi-C data obtained from Bickhard et al.[33]. Each haplotype assembly was scaffolded to chromosome-level with Hi-C data using YaHS. A Hi-C contact map was generated for the resulting scaffolds using Juicer pipeline, and manual curation was done based on the Hi-C contact map with JuiceBox, resulting in the final chromosome-level haplomes.

## Aligning repetitive and nonrepetitive sequences to haplomes

Repetitive sequences of one haplome were defined as regions in this haplome with a coverage of 27-mers from the other haplome to be equal to or greater than 2. Excluding gaps, any regions in one haplome with a coverage of 27-mers from the other haplome to be lower than 2 were defined as nonrepetitive. The repetitive sequences in haplome 2 were first detected using KAT (sect -E -F h2.fa h1.fa), and then the repetitive sequences were masked with Ns. Non-N sequences were extracted as nonrepetitive sequences using BEDTools and SeqKit, and these nonrepetitive sequences were aligned to haplome 1 using minimap2 (-ax asm20 −eqx). After mapping of nonrepetitive sequences, the nonrepetitive sequences in the original unmasked assembly were masked with Ns, and non-N sequences were extracted as repetitive sequences using BEDTools and SeqKit. The repetitive sequences were aligned to haplome 1 using minimap2 (-ax asm20 −eqx). Only primary alignments were kept, and the alignment files (in SAM format) generated by minimap2 were converted to BED format using BEDTools (bedtools bamtobed). The cigar string of the alignment was kept in the BED file, and the cigar string was parsed with a custom python script. Based on the cigar string parsing, for each primary alignment, the sequence identity was calculated as the number of matched bases divided by the total length of the alignment which includes matched bases, mis-matched bases, deletions, insertions and soft-clipped bases. The calculation of the alignment sequence identity and the visualization of the alignments were done with an R script (Fig. 4).

## Calculating repetitive and nonrepetitive sequence coverage

First, a bed file indicating the position of each repetitive and nonrepetitive sequence in each haplome was generated with the same method described above. Next, for every 1 Mb window in the pseudo-chromosome, the number of repetitive and nonrepetitive bases were counted using BEDTools (bedtools coverage), namely, how many bases in 1 Mb were covered by repetitive and nonrepetitive sequences, respectively. The coverage was visualized with an R script (Supplementary Fig. 11a).

## Genome sequencing

The ONT data of DH647 and Sikem were generated with the ONT library preparation protocol customized for ryegrass[16]. The WGS pair-end short read data of DH647 and Sikem were generated by Functional Genomics Center Zurich with Illumina NovaSeq. The PacBio HiFi data of Sikem were generated by Arizona Genomics Institute with PacBio Sequel II. The Hi-C data of DH647 and Sikem were generated with the in situ Hi-C protocol[17,46] and sequenced using the Illumina NovaSeq 6000 device at IPK, Gatersleben.

## Estimating the genome size and the level of heterozygosity

With the 21-mers counted by Jellyfish[68] from the WGS short read data, the genome size and the level of heterozygosity of DH647 and Sikem was estimated by GenomeScope2[69] (Supplementary Fig. 1).

## HapHiC for Sikem

HapHiC[7,14] was applied to the same unitig assembly of Sikem that was used for PhaseGrass with the same Hi-C data of Sikem, following the standard pipeline (https://github.com/zengxiaofei/HapHiC/tree/main). The code used for HapHiC is available at Github (https://github.com/Yutang-ETH/PhaseGrass_review).

## Reporting summary

Further information on research design is available in the Nature Portfolio Reporting Summary linked to this article.

# Data availability

All genome sequencing data produced in this work including ONT reads of DH647 and Sikem, PacBio HiFi reads of Sikem, Hi-C reads and WGS short reads of DH647 and Sikem are all available on NCBI under BioProject PRJNA1156640 and PRJNA1156633. The chromosome-level, haplotype-resolved assemblies of DH647 and Sikem are available on NCBI under BioProject PRJNA1147195 for DH647 h1, PRJNA1147194 for DH647 h2, PRJNA1147193 for Sikem h1, and PRJNA1147192 for Sikem h2. The red clover HEN17 haplotype-resolved assembly generated in this study is available at Zenodo [https://zenodo.org/records/15411223]. Source data are provided with this paper.

## Code availability

The PhaseGrass workflow is openly accessible at GitHub [https://github.com/Yutang-ETH/PhaseGrass]. All code used for evaluation and comparison analyses in this work is also available at GitHub [https://github.com/Yutang-ETH/PhaseGrass_publication] and [https://github.com/Yutang-ETH/PhaseGrass_review].

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

## Acknowledgements

This work was supported by the European Union's Horizon 2020 research and innovation program under the Marie Skłodowska-Curie grant agreement No 847585 - RESPONSE. We sincerely thank Verena Knorst from the Molecular Plant Breeding group at ETH Zurich for taking care of the plant material. We sincerely thank Manuela Knauft (IPK Gatersleben) for the in situ Hi-C library preparation and Ines Walde (IPK Gatersleben) for Hi-C sequencing. We appreciate valuable discussions on genome assembly and phasing with Jiawu Feng (IPK Gatersleben). We thank the Functional Genomics Centre Zurich (FGCZ) for providing the short-read sequencing data. We sincerely thank ISG-HEST at ETH Zurich for providing computational resources as well as their IT service for this work.

## Author contributions

Y.C., D.C., R.K. and B.S. conceived this study, for which D.C. and B.S. acquired funding. Y.C. developed PhaseGrass, completed the genome assembly, performed the evaluation and comparison analyses and drafted the manuscript. D.C. generated the ONT and PacBio HiFi data, supervised this work and edited the manuscript. R.K. and B.S. supervised this work and edited the manuscript. M.M. supervised assembly and phasing analyses and reviewed the manuscript. A.H. and N.S. generated the Hi-C data and reviewed the manuscript.

## Funding

## Competing interests

The authors declare no competing interests.
