## [Peer Review file · Nature Communications]

Chromosome-level haplotype-resolved assembly of highly heterozygous grass genomes with PhaseGrass

Corresponding Author: Professor Bruno Studer

Version 0:

Reviewer comments:

Reviewer #1

(Remarks to the Author)

This paper presents PhaseGrass, a novel end-to-end haplotype-resolved assembly workflow designed specifically for genomes with high heterozygosity. Using PhaseGrass, the authors generated haplotype-resolved assemblies for both *Lolium perenne* and *L. multiflorum*, and conducted a thorough evaluation of the results.

While the authors make a valuable contribution to the community, I have several concerns and suggestions that I believe should be addressed or discussed.

Major specific comments:

1. The authors state (lines 52–53): “This is potentially due to some highly divergent DNA sequences between haplotypes that cannot be paired to form ‘bubbles’ in a phased assembly graph.” However, this statement is inaccurate. Existing haplotype-resolved assemblers—such as hifiasm, Verkko, and Falcon-Phase—employ an all-vs-all alignment step specifically to identify homologous regions with high heterozygosity that cannot be organized into bubble structures.
2. DipGrass requires short-read data for SNP calling. I was wondering why it cannot directly perform SNP calling using long-read data?
3. When assembling DH647, why is the assembly of another inbred sample used as the reference, rather than using the primary assembly of DH647 itself? In this case, PhaseGrass is a reference-guided assembly algorithm rather than a true de novo assembler.
4. In Figure 2b, PhaseGrass phases more reads than WhatsHap. It would be helpful to clarify the reason for this difference. Is it because PhaseGrass uses additional short-read data, or do PhaseGrass and WhatsHap rely on different reference assemblies? It is unclear whether WhatsHap also uses the haploid assembly of Kyuss as the reference.
5. The contig N50 of the assemblies produced by PhaseGrass is substantially smaller compared to those generated by hifiasm. To improve assembly contiguity, the authors could consider using the -3/-4 options in hifiasm to incorporate phasing information into the assembly process.
6. It would be better to show HapHiC results for Sikem with PacBio HiFi reads.
7. In Step 2 of PhaseGrass, PurgeGrass leverages all-vs-all alignment, micro-synteny, and BUSCO genes to pair allelic contigs and remove redundancies. Since most existing tools rely solely on all-vs-all alignment, I was curious whether the incorporation of micro-synteny and BUSCO genes significantly contributes to the improved performance of PhaseGrass. However, relying on BUSCO genes may introduce limitations for species with naturally high levels of gene duplication.

(Remarks on code availability)

PhaseGrass was able to run successfully during our test.

Reviewer #2

(Remarks to the Author)

Until now, it was challenging to generate balanced haplotype resolved assemblies of highly heterozygous genomes. The manuscript reports on a new assembling, scaffolding and phasing workflow, combining reference-based phasing with haplotype-specific k-mers. The biggest advantage of the workflow described in the manuscript is no need for parental data, use of both high fidelity PacBio and more error prone ONT data, and creation of balanced haplotypes for the heterozygous

species. The authors used the workflow to assemble two species of highly heterozygous ryegrass, *Lolium* spp., for which a frequently used graph-based phasing using Hifiasm resulted in the imbalanced haplotypes. Besides the assemblies of *L. multiflorum* and *L. perenne*, which contain a large proportion of repeats (~ 70%), the assembly of a genotype of *Trifolium pratense*, represented by a smaller genome and a low proportion of repeats (~ 20%) was created. As the balanced haplotypes of *T. pratense* were obtained using the Hifiasm, the authors also investigated why Hifiasm failed in creating of balanced haplotypes of *Lolium* species.

The manuscript is very well-written, the workflow and other results are clearly described and completed on sufficient amount of Figures and Tables.

Even though the new workflow can only be applied to diploid species with heterozygosity uniformly distributed through the genomes, it provides an important tool for the routine creation of haplotype-resolved genomes, as well as for the re-analysis of the already existing assemblies represented by the imbalanced haplotypes.

I have only minor comments on the manuscript.

1) Line 84, it is recommended „For genomes with large chromosomes, we recommend using TRITEX for scaffolding.“ From cytogenetic point of view, even species of *Lolium* have chromosomes at moderate/large sizes (N50 ~ 300 Mb), in comparison to many other plant species (e.g. *Arabidopsis*, *Musa* spp, *Solanum* spp., ...). Is it possible to recommend the length of the chromosomes for which TRITEX should be used more precisely?

2) The authors mentioned that there is no need of parental reference assembly to obtain phased genome assemblies of highly heterozygous genomes.

On the other hand, on Line 107 – 109, they mentioned that in case of *L. perenne* „We used the high quality, chromosome-level haploid assembly of Kyuss (contig N50 of 120 Mb), a double haploid progeny derived from another culture of DH674, as the reference to phase DH647.“ This can be clarified. Why, in this case, an already existing reference genome was used to phase heterozygous diploid species DH647?

3) Line 150, and Line 159, the authors refer to Supplementary Fig.3, but I guess that it should be Extended data Fig. 4.

(Remarks on code availability)

The README files are available for every single step of the pipeline.
Detailed instructions for installing and usage are available.

Unfortunately, I did not have time to install whole PhaseGrass pipeline and run the code.

Version 1:

Reviewer comments:

Reviewer #1

(Remarks to the Author)

The reviewers have addressed most of my comments, so I believe the manuscript is ready for publication.

I have only one suggestion regarding my comment #2. In the authors' response, their main reason for using short reads instead of long reads for SNP calling is that short reads are easier to align. However, it may be possible to fragment long reads into smaller pieces, similar to short reads. In that case, the advantages of long reads for SNP calling may not be necessary.

(Remarks on code availability)

The code is ok to run.

Reviewer #2

(Remarks to the Author)

Thank you for answering/clarifying all the comments.

(Remarks on code availability)

Reviewer #1 (Remarks to the Author):

This paper presents PhaseGrass, a novel end-to-end haplotype-resolved assembly workflow designed specifically for genomes with high heterozygosity. Using PhaseGrass, the authors generated haplotype-resolved assemblies for both *Lolium perenne* and *L. multiflorum*, and conducted a thorough evaluation of the results.

While the authors make a valuable contribution to the community, I have several concerns and suggestions that I believe should be addressed or discussed.

Major specific comments:

1. The authors state (lines 52–53): “This is potentially due to some highly divergent DNA sequences between haplotypes that cannot be paired to form ‘bubbles’ in a phased assembly graph.” However, this statement is inaccurate. Existing haplotype-resolved assemblers—such as hifiasm, Verkko, and Falcon-Phase—employ an all-vs-all alignment step specifically to identify homologous regions with high heterozygosity that cannot be organized into bubble structures.

We fully agree and have revised the writing to be more precise, based on the following technical rationale of the above-mentioned assemblers (which follow the overlap–layout–consensus framework during assembly):

- The overlap step is where the all-vs-all read alignment is done. Reads from haplotypes at genomic regions with moderate level of heterozygosity will be aligned, while reads from haplotypes at highly heterozygous regions exceeding the sequence divergence threshold of the aligner will not be aligned (see Supplementary Figure 10 of falcon unzip, given below, available under <https://doi.org/10.1038/nmeth.4035>).

Supplementary Figure 10

General schematic about how different levels of heterozygosity can affect the contig layout.

- Then, the layout step builds the assembly graphs based on the all-vs-all alignment. Reads from haplotypes that are aligned (green, figure above) will form bubbles in a graph, while reads from two haplotypes that fail to be aligned (orange, figure above) will be split to form different independent graphs. This step does haplotype separation.
- Finally, the consensus step outputs the contigs based on the graphs. The two sides (up and down sides) of the bubbles in a graph can be assigned to different haplotypes with the help of external phasing information, such as a trio or Hi-C data. This step does haplotype partition. If a graph (with bubbles) spans the whole chromosome, then – ideally – it will result in two chromosome-level haplotypes, namely the h1 and h2 haplotype assemblies generated by hifiasm. However, if two haplotypes could not be aligned and have been split into two graphs in the layout step, then the original haplotype of these two graphs could not be determined since they don't form bubbles. These two graphs will be assigned to the same haplotype, namely, they will both be considered as primary contigs or they will be assigned to the same haplotype assembly, resulting in imbalanced haplotype assemblies. For example, in a diploid genome with two pairs of homologous chromosomes (where A pairs with B, and C pairs with D): A and B can be overlapped or aligned due to relatively lower heterozygosity, and C and D could not be aligned due to high heterozygosity. A and B will form one graph with bubbles inside, but C and D will form two graphs, with one representing C and the other representing D. The assembler will

correctly partition A to the haplotype 1 assembly and B to the haplotype 2 assembly, while C and D probably will be partitioned together to either haplotype 1 or 2 assembly, leading to imbalanced haplotype assemblies.

Haplotype separation is not a problem, and the higher the level of heterozygosity, the easier the haplotype separation (which is the case for outbreeding forage grasses such as ryegrasses). However, since hifiasm relies on the bubbles in the assembly graph to partition haplotypes, haplotype partition would be a problem if two haplotypes could not be aligned to form one graph with bubbles but two independent graphs.

We have improved the sentence from line 52 to 55 as follows to make it clearer:

“This is potentially due to some highly divergent DNA sequences between haplotypes that fail to be aligned during the all-vs-all alignment step of assembly which leads to two independent assembly graphs that are subsequently partitioned to the same haplome. Graph-based phasing needs DNA sequences to be aligned to form one graph with “bubbles”⁴ so that the two sides of the bubbles (alleles or haplotigs) can be partitioned to different haplotypes.”

2. DipGrass requires short-read data for SNP calling. I was wondering why it cannot directly perform SNP calling using long-read data?

We greatly appreciate this insightful question – basically, it is because of reference bias: When aligning reads to a haploid genome assembly (reads and assembly from the same genome), short reads are easier to be aligned to highly heterozygous regions compared to long reads. Thus, SNPs can be called from short-read alignments. Long reads that do not share the same haplotype as the assembly might not be aligned, leaving only long reads from the same haplotype as the assembly aligned. In this case, no variation can be detected. In Extended Data Fig. 1, we have illustrated reference bias and, in order to further clarify, we added the following sentences to the Discussion:

“Given the effect of reference bias, calling SNPs from the alignment of short reads is advantageous for reference-based phasing. This is because, if only long reads that share the same haplotype as the reference can be aligned to the assembly, then no variation between those long reads and the assembly would be detectable.”

3. When assembling DH647, why is the assembly of another inbred sample used as the reference, rather than using the primary assembly of DH647 itself? In this case, PhaseGrass is a reference-guided assembly algorithm rather than a true de novo assembler.

PhaseGrass does de novo assembly, but it does reference-based phasing based on the de novo primary assembly. As the genotype Kyuss is a doubled haploid progeny of DH647 (stated in line 107), Kyuss served as the primary assembly of DH647 for

reference-based phasing (not the assembly) to binning ONT reads of DH647 to different haplotypes. Binned reads were separately assembled subsequently de novo to reconstruct each haplome. We could generate a primary assembly for DH647 as the case of Sikem, but since Kyuss has a very high quality, we decided to use it.

Furthermore, Kyuss also served as a benchmark for phasing: We used it to evaluate the phasing quality of PhaseGrass, since we know each chromosome in Kyuss is only from one haplotype of DH647 with only one or two haplotype switches due to recombination. Phasing DH647 with Kyuss allowed direct comparison of the phase between DH647 haplotypes and Kyuss.

We have improved line 107-109 in the main text as below to make the context clearer:

“As a high-quality, chromosome-level haploid assembly of DH647 was already available (Kyuss³⁰, a doubled haploid progeny derived from anther culture of DH647³¹), we directly used this assembly for referenced-based phasing for DH647. This Kyuss assembly also served as a benchmark for phasing, allowing to evaluate the phasing quality of PhaseGrass by comparing the phase between DH647 haplotypes (generated by PhaseGrass) and Kyuss.”

4. In Figure 2b, PhaseGrass phases more reads than WhatsHap. It would be helpful to clarify the reason for this difference. Is it because PhaseGrass uses additional short-read data, or do PhaseGrass and WhatsHap rely on different reference assemblies? It is unclear whether WhatsHap also uses the haploid assembly of Kyuss as the reference.

WhatsHap, which conducts reference-based phasing, is the core component in DipGrass, the reference-based phasing pipeline in PhaseGrass. In the case of DH647, since PhaseGrass used WhatsHap for reference-based phasing, Kyuss is the only reference used for both PhaseGrass and WhatsHap. PhaseGrass outputs the reads partitioned to haplotypes by WhatsHap anyway but does not use these reads for downstream steps. The reason why more reads were assigned to haplotypes for PhaseGrass than WhatsHap is the step after reference-based phasing, SortGrass, which uses haplotype-specific k-mers to bin reads to different haplotypes. WhatsHap relies on long-read alignment to partition reads. In the presence of reference bias, many reads could not be aligned or inaccurately aligned to the assembly. Thus, these reads could not be partitioned to any haplotypes by WhatsHap. While binning long reads to different haplotypes with haplotype-specific k-mers extracted from the chromosome-level haplotypes generated by the reference-based phasing step, SortGrass avoids reference bias, thus having the potential to bin more reads to haplotypes.

For clarification, we added following sentences in the Discussion:

“For highly divergent regions with large presence and absence variations, reference bias might lead only long reads sharing the same haplotype with the reference to be aligned to the assembly. Thus, less reads could be assigned to haplotypes by WhatsHap. In contrast, it is easier to match short k-mers (21 bp) with sequences from highly divergent regions. Thus, k-mers can improve long read or unitig partition.”

5. The contig N50 of the assemblies produced by PhaseGrass is substantially smaller compared to those generated by hifiasm. To improve assembly contiguity, the authors could consider using the -3/-4 options in hifiasm to incorporate phasing information into the assembly process.

PhaseGrass partitions unitigs not contigs generated by hifiasm to different haplotypes. This is similar to HapHiC, which clusters unitigs to scaffold groups. Contigs generated by hifiasm are normally longer than unitigs but may have phase switch errors. A unitig is truly representing one haplotype. Here, relating back to our explanation to point 1, PhaseGrass tries to avoid the haplotype partition problem that hifiasm has for highly heterozygous genomes. Hifiasm builds the graph first and then partitions the haplotype. The -3/-4 options under the trio binning mode of hifiasm come into effect after the graphs have been built and help in partitioning the two sides of bubbles in a graph to different haplotypes. They would not help to partition two graphs from two haplotypes that could not be aligned. Therefore, using these options will lead to imbalanced haplotypes.

6. It would be better to show HapHiC results for Sikem with PacBio HiFi reads.

We highly appreciate this point, as it raised our curiosity about the performance of HapHiC. We successfully ran HapHiC by following the instructions provided in the authors' GitHub repository, and our implementation of HapHiC is made available at: https://github.com/Yutang-ETH/PhaseGrass_review. The results of HapHiC have been incorporated into the manuscript, including an additional paragraph in the corresponding Results section, as well as Supplementary Fig. 4 and Supplementary Table 6. Please see the added paragraph below:

“HapHiC was applied to Sikem for comparison with PhaseGrass. HapHiC resulted in 14 scaffolds (groups), corresponding to the 14 haplotypes (Supplementary Fig. 4a, Supplementary Table 6). Notably, these scaffolds were not partitioned to haplotypes, and homologous chromosome pairs were not identified. Compared with HapHiC, PhaseGrass anchored 262 Mb more unitigs to pseudo-chromosomes, showing a higher scaffold N50 (292 Mb vs 263 Mb, Supplementary Fig. 4b).”

7. In Step 2 of PhaseGrass, PurgeGrass leverages all-vs-all alignment, micro-synteny, and BUSCO genes to pair allelic contigs and remove redundancies.

Since most existing tools rely solely on all-vs-all alignment, I was curious whether the incorporation of micro-synteny and BUSCO genes significantly contributes to the improved performance of PhaseGrass. However, relying on BUSCO genes may introduce limitations for species with naturally high levels of gene duplication.

Thank you very much for pointing this out. In a pilot study with another *Lolium multiflorum* genome, which is where the prototype of PurgeGrass was developed, we have compared the performance between PurgeGrass and other tools relying solely on all-vs-all alignment (please see Supplementary Table 4 of Chen et al. 2025, given below). Haplotig purging with all-vs-all alignment-based tools like Purge Haplotigs and purge_dups resulted in around 40% duplicated BUSCO genes, while PurgeGrass resulted in much less (around 10%) duplicated BUSCO genes. Notably, the dramatic decrease of duplicated BUSCO genes with PurgeGrass did not result in a huge loss in the total BUSCO genes. Relying on BUSCO for purging might cause loss of some completeness, but compared to the under purge of all-vs-all alignment-based tools, it is acceptable as long as the loss is in an acceptable range. As we wrote in the Methods, “A duplicated BUSCO score lower than 10% together with a small loss in total BUSCO score (<5%) indicates a successful purge of haplotigs”, there is always a tradeoff between under purge and over purge. For clarification, we referred to this pilot study in the Methods, with the following sentence:

“Based on a pilot study with another Lolium multiflorum genotype from a Swiss cultivar ‘Rabiosa’ (hereafter referred to as Rabiosa), we have seen the potential of purging more redundant haplotigs using micro-synteny and BUSCO genes⁷.”

Assemblies	Complete and single-copy	Complete and duplicated	Fragmented	Missing	Total complete
Flye polished contigs	26.60%	72.70%	0.20%	0.50%	99.30%
Flye polished contigs purged by Purge Haplotigs	58.90%	38.30%	0.50%	2.30%	97.20%
Flye polished contigs purged by purge_dups	58.70%	36.40%	0.60%	4.30%	95.10%
Flye polished contigs purged by PurgeGrass	85.80%	8.40%	0.90%	4.90%	94.20%
Rabiosa v1	88.20%	5.80%	0.80%	5.20%	94.00%
Canu polished contigs	22.50%	76.00%	0.20%	1.30%	98.50%
Canu polished contigs purged by Purge Haplotigs	55.00%	40.80%	0.40%	3.80%	95.80%
Canu polished contigs purged by PurgeGrass	84.10%	11.00%	0.20%	4.70%	95.10%
Rabiosa v2	89.00%	5.60%	0.40%	5.00%	94.60%
Rabiosa h1	83.60%	6.10%	0.30%	10.00%	89.70%
Rabiosa h2	83.60%	5.40%	0.60%	10.40%	89.00%
Rabiosa dip	23.70%	72.60%	0.10%	3.60%	96.30%

Chen, Y. et al. Chromosome-level haplotype-resolved genome assembly provides insights into the highly heterozygous genome of Italian ryegrass (*Lolium multiflorum* Lam.). The Plant Genome 18, e70079 (2025).

Reviewer #2 (Remarks to the Author):

Until now, it was challenging to generate balanced haplotype resolved assemblies of highly heterozygous genomes. The manuscript reports on a new assembling, scaffolding and phasing workflow, combining reference-based phasing with haplotype-specific k-mers. The biggest advantage of the workflow described in the manuscript is no need for parental data, use of both high fidelity PacBio and more error prone ONT data, and creation of balanced haplotypes for the heterozygous species. The authors used the workflow to assemble two species of highly heterozygous ryegrass, *Lolium* spp., for which a frequently used graph-based phasing using Hifiasm resulted in the imbalanced haplotypes. Besides the assemblies of *L. multiflorum* and *L. perenne*, which contain a large proportion of repeats (~ 70%), the assembly of a genotype of *Trifolium pratense*, represented by a smaller genome and a low proportion of repeats (~ 20%) was created. As the balanced haplotypes of *T. pratense* were obtained using the Hifiasm, the authors also investigated why Hifiasm failed in creating of balanced haplotypes of *Lolium* species.

The manuscript is very well-written, the workflow and other results are clearly described and completed on sufficient amount of Figures and Tables.

Even though the new workflow can only be applied to diploid species with heterozygosity uniformly distributed through the genomes, it provides an important tool for the routine creation of haplotype-resolved genomes, as well as for the re-analysis of the already existing assemblies represented by the imbalanced haplotypes.

Thank you very much for your appreciation of our work!

I have only minor comments on the manuscript.

1) Line 84, it is recommended „For genomes with large chromosomes, we recommend using TRITEX for scaffolding. “From cytogenetic point of view, even species of *Lolium* have chromosomes at moderate/large sizes (N50 ~ 300 Mb), in comparison to many other plant species (e.g. *Arabidopsis*, *Musa* spp, *Solanum* spp., ...). Is it possible to recommend the length of the chromosomes for which TRITEX should be used more precisely?

We highly appreciate this suggestion. TRITEX is particularly beneficial for genomes with large chromosomes. Successful applications of TRITEX has been seen in genomes with chromosomes longer than 500 Mb, like barley (<https://doi.org/10.1038/s41586-025-09270-x>), wheat (<https://doi.org/10.1038/s41586-020-2961-x>), rye (<https://doi.org/10.1038/s41588-021-00807-0>), oat (<https://doi.org/10.1038/s41586-022-04732-y>) and faba bean (<https://doi.org/10.1038/s41586-023-05791-5>). It is hard to draw a threshold of chromosome length for TRITEX, but if the chromosome size of the target genome is similar to our ryegrass genomes, for example over 200 Mb, then it might be a good

idea to consider TRITEX. However, TRITEX is also applicable for genomes with smaller chromosomes. Also notably, if contigs are long enough, for example N50 around 30 Mb according to Jiao et al. (<https://doi.org/10.1038/s41586-024-08277-0>), other scaffolding pipelines might also be helpful for large chromosomes.

We have revised the sentence at line 84 as follows:

“For genomes with large chromosomes, we recommend using TRITEX²³ for scaffolding, although any other effective scaffolding tool may also be applied”.

2) The authors mentioned that there is no need of parental reference assembly to obtain phased genome assemblies of highly heterozygous genomes.

On the other hand, on Line 107 – 109, they mentioned that in case of *L. perenne* „We used the high quality, chromosome-level haploid assembly of Kyuss (contgi N50 of 120 Mb), a double haploid progeny derived from another culture of DH674, as the reference to phase DH647.“ This can be clarified. Why, in this case, an already existing reference genome was used to phase heterozygous diploid species DH647?

Thank you very much for raising this point, which was also mentioned by Reviewer 1 and has been clarified accordingly. Theoretically, we could have generated a primary assembly for DH647 (as was done for Sikem). However, we chose to use the already existing Kyuss genome because of its very high quality and its suitability for benchmarking the phasing: Phasing DH647 with Kyuss allowed direct comparison of the phase between DH647 haplotypes and Kyuss, thereby allowing us to evaluate and visualize the phasing quality of PhaseGrass.

We have improved line 107-109 as below to make the context clearer:

“As a high-quality, chromosome-level haploid assembly of DH647 was already available (Kyuss³⁰, a doubled haploid progeny derived from another culture of DH647³¹), we directly used this assembly for referenced-based phasing for DH647. This Kyuss assembly also served as a benchmark for phasing, allowing to evaluate the phasing quality of PhaseGrass by comparing the phase between DH647 haplotypes (generated by PhaseGrass) and Kyuss.”

3) Line 150, and Line 159, the authors refer to Supplementary Fig.3, but I guess that it should be Extended data Fig. 4.

We greatly appreciate the careful reading of our manuscript. However, we believe that what is presented in the manuscript is correct: Line 150 refers to the chromosome-level unphased haploid assembly of Sikem, which is correctly linked to Supplementary Fig. 3, where the Hi-C contact map of the haploid assembly was shown. Line 159 explains why there are more small phase blocks resulting from PacBio HiFi reads than

ONT reads. This is correctly supported by Extended Data Fig. 3, where the IGV screen shots showed that longer ONT reads resulted in more contiguous phasing. Extended Data Fig. 4 shows the diploid Hi-C contact map of the haplotype-resolved assembly of Sikem, and other quality evaluation analyses based on the haplotype-resolved assembly.

Reviewer #2 (Remarks on code availability):

The README files are available for every single step of the pipeline.

Detailed instructions for installing and usage are available.

Unfortunately, I did not have time to install whole PhaseGrass pipeline and run the code.